# Impact of Phantom Size on Low-Energy Virtual Monoenergetic Images of Three Dual-Energy CT Platforms

**DOI:** 10.3390/diagnostics13193039

**Published:** 2023-09-25

**Authors:** Joël Greffier, Claire Van Ngoc Ty, Isabelle Fitton, Julien Frandon, Jean-Paul Beregi, Djamel Dabli

**Affiliations:** 1IMAGINE UR UM 103, Department of Medical Imaging, Nimes University Hospital, Montpellier University, 30029 Nimes, France; julien.frandon@chu-nimes.fr (J.F.); jean.paul.beregi@chu-nimes.fr (J.-P.B.); djamel.dabli@chu-nimes.fr (D.D.); 2Department of Radiology, Assistance Publique Hôpitaux de Paris, Hôpital Européen Georges Pompidou, Université de Paris, 75015 Paris, France; claire.vanngocty@aphp.fr (C.V.N.T.); isabelle.fitton@aphp.fr (I.F.)

**Keywords:** dual-energy, multidetector computed tomography, task-based image quality assessment, split-filter

## Abstract

The purpose of this study was to compare the quality of low-energy virtual monoenergetic images (VMIs) obtained with three Dual-Energy CT (DECT) platforms according to the phantom diameter. Three sections of the Mercury Phantom 4.0 were scanned on two generations of split-filter CTs (SFCT-1st and SFCT-2nd) and on one Dual-source CT (DSCT). The noise power spectrum (NPS), task-based transfer function (TTF), and detectability index (d’) were assessed on VMIs from 40 to 70 keV. The highest noise magnitude values were found with SFCT-1st and noise magnitude was higher with DSCT than with SFCT-2nd for 26 cm (10.2% ± 1.3%) and 31 cm (7.0% ± 2.5%), and the opposite for 36 cm (−4.2% ± 2.5%). The highest average NPS spatial frequencies and TTF values at 50% (f_50_) values were found with DSCT. For all energy levels, the f_50_ values were higher with SFCT-2nd than SFCT-1st for 26 cm (3.2% ± 0.4%) and the opposite for 31 cm (−6.9% ± 0.5%) and 36 cm (−5.6% ± 0.7%). The lowest d’ values were found with SFCT-1st. For all energy levels, the d’ values were lower with DSCT than with SFCT-2nd for 26 cm (−6.2% ± 0.7%), similar for 31 cm (−0.3% ± 1.9%) and higher for 36 cm (5.4% ± 2.7%). In conclusion, compared to SFCT-1st, SFCT-2nd exhibited a lower noise magnitude and higher detectability values. Compared with DSCT, SFCT-2nd had a lower noise magnitude and higher detectability for the 26 cm, but the opposite was true for the 36 cm.

## 1. Introduction

In recent years, numerous technological developments and innovations have emerged in CT imaging, leading to the better radiological management of patients [1,2,3,4,5,6,7,8,9,10,11]. Among them, dual-energy spectral imaging has emerged to improve the detection and characterization of lesions, particularly abdominal lesions [12,13,14,15]. Its basic principle is to acquire or detect two photon spectra at low- and high-energy levels to help identify the tissue attenuation coefficients. To achieve this, various dual-energy CT (DECT) platforms have been developed [1,2,16,17,18].

Among the various DECT platforms, one manufacturer has developed two platforms for obtaining two-photon spectra during acquisitions: the dual-source CT (DSCT) platform and the split-filter CT (SFCT) platform [1,16]. For the DSCT platform, two X-ray tube/detector pairs (95° offset for 3rd generation) are used to acquire the image datasets. The X-ray tube “A” uses low kVp (70, 80, 90, and 100 kVp) and the other X-ray tube “B” uses high kVp with (Sn150 kVp) or without a tin filter (140 kVp). With this platform, spectral acquisitions can be made with five pairs of kVps, and the choice of these pairs depends on the clinical indication for the CT examination and the patient’s morphology. For the SFCT platform, the photon beam is split into two-energy spectra using two filters placed at the outlet of the X-ray tube used. The gold filter (Au) is used for the low-energy spectrum and the tin filter (Sn) for the high-energy spectrum. In the first-generation platform (SFCT-1st), the gold filter is 0.05 mm thick and the tin filter, 0.6 mm. In the second-generation platform (SFCT-2nd), the thickness of the gold filter has been increased to 0.07 mm, and the thickness of the tin filter has been increased to 0.70 mm. For the SFCT-1st, only 120 kVp can be used to acquire the spectral image datasets, whereas 120 and 140 kVp can be used for the SFCT-2nd.

Two studies have highlighted the differences in the spectral performance between SFCT-1st and DSCT [16,17]. Using two X-ray tubes provides good spectral separation for DSCT. Otherwise, with SFCT, spectral separation is more limited, due to the overlap between the low- and high-energy spectra generated by the two filters. These differences in the spectral performance translate into greater image noise and poorer lesion detectability on the virtual monoenergetic images (VMIs) at low energy levels [16,17]. Only one study has evaluated the impact of modifying the thickness of the two filters on the spectral performance of SFCT-2nd, compared with SFCT-1st [19]. The authors demonstrated that the modifications in the thickness to the gold and the tin filters of SFCT-2nd improved spectral separation and also the spectral performance (especially, the lowest values of noise magnitude and the highest values of the detectability indexes) compared to SFCT-1st for the low-keV VMIs. All three studies were carried out using a phantom with a fixed morphology. In addition, the spectral image quality is also influenced by patient morphology and the associated dose level. Euler et al. evaluated the impact of the phantom diameter and dose level on the spectral performance of the DSCT platform [20]. However, no similar studies have been conducted on the two generations of SFCT platforms.

The purpose of our study was to compare the spectral performances of three DECT platforms according to the phantom diameter. To achieve this goal, the spectral performances of the three DECT platforms were assessed by carrying out a task-based image quality assessment on the VMIs at low energy levels using the three largest sections of an image quality phantom.

## 2. Materials and Methods

### 2.1. Phantom

The three largest sections (26, 31, and 36 cm diameter) of the Mercury v4.0 phantom (Sun Nuclear, Melbourne, FL, USA) were used to perform the task-based image quality assessment (Figure 1A). These three sections have diameters corresponding to the abdomen of patients with a body mass index of 18, 27, and 36 kg/m^2^, respectively [21]. Each section of the phantom is made up of a homogeneous zone with which the Noise Power Spectrum can be calculated (NPS; Figure 1B) and a section with five inserts for calculating the Task-based Transfer Function (TTF), particularly on the iodine insert at 10 mg/mL (Figure 1C).

### 2.2. CT Scanners and Scanning Protocol

Acquisitions were performed on three Siemens Healthineers (Forchheim, Germany) DECT systems, the SOMATOM Force (DSCT), and two generations of SFCT: the SOMATOM Edge (SFCT-1st) and the SOMATOM X.Cite (SFCT-2nd).

For each CT system, the acquisition and reconstruction parameters usually used in clinical practice for abdomen DECT examination were selected (Table 1). For each CT system, the automatic tube current system was disabled and the tube currents were set to obtain a volume CT dose index (CTDI_vol_) of 12.0, 9.1, and 6.9 mGy for the 36 cm, 31 cm, and 26 cm diameter phantoms, respectively. These dose levels correspond to those usually used on these three CT systems for DECT abdomen–pelvis examinations, according to the patient’s morphology. Each acquisition was repeated 10 times.

The raw data were reconstructed using Level 3 of the iterative reconstruction algorithm ADMIRE (ADvanced Modeled Iterative REconstruction) on each CT system. For all systems, the quantitative reconstruction kernel (Qr40), a slice thickness of 1 mm, and a 380 mm field of view were used. For each acquisition, the VMIs for the four lowest energy levels (40/50/60/70 keV) were reconstructed on Syngo.via software (VB60A_HF03) with the specific Monoenergetic Plus application.

### 2.3. Task-Based Image Quality Assessment on VMIs

A task-based image quality assessment was performed on the VMIs using the iQMetrix-CT software (v1.1) developed by the French Society of Medical Physicists [22]. For each CT system and each energy level, the Noise Power Spectrum (NPS), Task-based Transfer Function (TTF), and detectability index were computed based on all the data from the 10 acquisitions. A single calculation for each of these metrics was carried out per energy level.

#### 2.3.1. Noise Power Spectrum

For each energy level, the NPS was computed in 180 consecutive axial slices (18 slices for each of the 10 acquisitions) by placing four square ROIs (Figure 1B). The sizes of the four square ROIs were adjusted according to the phantom’s sections: 80 × 80 pixels for the 26 cm, and 104 × 104 pixels for 31 cm and 120 × 120 pixels for the 36 cm diameters.

To quantify the changes in noise magnitude, the square root of the area under the NPS1D curve (HU) and the magnitude of the NPS1D peak (HU^2^·mm^2^) were measured. To quantify the changes in noise texture, the average spatial frequency (f_av_, mm^−1^) of the NPS curve and the spatial frequency of the NPS peak(s) (f_peak_, mm^−1^) were measured.

#### 2.3.2. Task-Based Transfer Function

For each VMI, the TTF was computed on iodine inserts at 10 mg/mL (Figure 1C) using the circular edge technique [23]. To minimize the image noise effect, the TTF was computed from 160 consecutive axial slices (16 slices for each of the 10 acquisitions).

The TTF values at 50% (f_50_, mm^−1^) were used to quantify the changes in spatial resolution.

#### 2.3.3. Detectability Index

A non-prewhitening observer model with an eye filter (d’_NPWE_) was used to calculate the detectability index (d’) for one 10 mm diameter clinical task approaching the contrast of enhanced vascular or strongly enhancing parenchymal structures like, for example, a hepatocellular carcinoma [20]. Based on the variations in contrast between the phantom’s background material and the iodine insert according to the energy level, the contrast was set at 1260, 820, 550, and 390 HU for the energy levels ranging from 40 to 70 keV.

The shape of the signal was circular and the contrast profile was Gaussian [24]. The interpretation conditions for calculating the d’ were a zoom factor of 1.5, a 500 mm viewing distance, and the Eckstein visual function [25].

## 3. Results

The TTF curves obtained for each CT system, each phantom’s diameter, and each keV are presented in the Appendix A. The axial images of the iodine insert at 10 mg/mL obtained with the 31 cm diameter for the three CT systems according to the energy level are depicted in the Appendix B. In the Results section, the values are expressed as means ± standard deviations (SD).

### 3.1. Noise Power Spectrum

#### 3.1.1. Noise Magnitude

For all CT systems and all diameters of the phantoms, the noise magnitude decreased as the energy level increased (Figure 2A). From 40 to 70 keV, the noise magnitude decreased on average for all diameters of the phantoms by −59.0% ± 3.3% for SFCT-1st, −57.1% ± 1.6% for SFCT-2nd, and −55.4% ± 0.8% for DSCT.

For all CT systems and all energy levels, the noise magnitude increased as the phantoms’ diameters increased. From the diameter of 26 cm to 31 cm, the noise magnitude increased on average for all energy levels by 40.5% ± 3.2% for SFCT-1st, 36.3% ± 2.7% for SFCT-2nd, and 32.3% ± 1.0% for DSCT and from 31 cm to 36 cm, 41.4% ± 6.7%, 31.6% ± 1.7%, and 17.8% ± 1.0%, respectively.

Whatever the phantom diameter and energy level, the highest noise magnitude values were found with SFCT-1st. For all energy levels, the noise magnitude was higher with DSCT than with SFCT-2nd for the diameters of 26 cm (10.2% ± 1.3%) and 31 cm (7.0% ± 2.5%), and the opposite for the diameter of 36 cm (−4.2% ± 2.5%).

#### 3.1.2. Noise Texture

For all CT systems and all phantom diameters, the average NPS spatial frequencies (f_av_) increased as the energy level increased (Figure 2B). From 40 to 70 keV, the f_av_ values increased on average for both SFCTs by 20.1% ± 0.3% for 26 cm and 23.5% ± 1.9% for 31 cm. For the 36 cm diameter phantom, the variations in the f_av_ values were more marked for SFCT-1st (from 0.13 to 0.22 mm^−1^) than for SFCT-2nd (from 0.17 to 0.22 mm^−1^). For DSCT, the f_av_ values increased for all phantom diameters by 11.0% ± 2.7% on average from 40 to 70 keV.

For all CT systems and all energy levels, the f_av_ values decreased as the phantoms’ diameter increased. From 26 cm to 31 cm in diameter, the noise magnitude increased on average for all energy levels by −5.0% ± 1.6% for SFCT-1st, −5.1% ± 0.7% for SFCT-2nd, and −4.1% ± 0.4% for DSCT, and from 31 cm to 36 cm, −21.3% ± 10.1%, −11.0% ± 3.1%, and −12.5% ± 1.4%, respectively.

Whatever the phantom diameter and energy level, the highest f_av_ values were found with DSCT. For all energy levels, the f_av_ values were similar with SFCT-1st and SFCT-2nd for the diameters of 26 cm and 31cm but for 36 cm, higher f_av_ values were found with SFCT-2nd, and all the more so at low keV levels.

With both SFCT systems and all phantom diameters, two NPS peaks were present on the curves for 40, 50, and 60 keV (and 70 keV only for SFCT-1st and the 36 cm diameter): one at a spatial frequency of 0.02 to 0.04 mm^−1^ and another at 0.14 to 0.17 mm^−1^ (Figure 3 and Table 2). With DSCT, two NPS peaks were present at 40 keV for all diameters and at 50 and 60 keV only for the 36 cm diameter phantom. The magnitude of the low frequency NPS peak was higher than the high frequency NPS peak for all systems, but the magnitude difference between these two peaks decreased as the energy levels increased.

### 3.2. Task-Based Transfer Function

For all CT systems and all diameters of the phantoms, the TTF values at 50% (f_50_) increased as the energy level increased (Figure 4 and Appendix A). From 40 to 70 keV, the f_50_ values increased on average for all the diameters of phantoms by 7.6% ± 0.6% for SFCT-1st, 6.3% ± 0.3% for SFCT-2nd, and 5.2% ± 0.5% for DSCT.

For all CT systems and all energy levels, the f_50_ values decreased as the diameter phantoms increased. From the diameter of 26 cm to 31 cm, the f_50_ values decreased on average for all energy levels by −3.9% ± 0.4% for SFCT-1st, −13.2% ± 0.2% for SFCT-2nd, and −7.6% ± 0.3% for DSCT, and from 31 cm to 36 cm, −6.4% ± 0.2%, −5.1% ± 0.1%, and −11.9% ± 0.1%, respectively.

Whatever the phantom diameter and energy level, the highest f_50_ values were found with DSCT, except at 70 keV for the 36 cm diameter phantom. For all energy levels, the f_50_ values were higher with the SFCT-2nd platform than with the SFCT-1st platform for the 26 cm diameter (3.2% ± 0.4%) and the opposite for the 31 cm diameter (−6.9% ± 0.5%) and 36 cm (−5.6% ± 0.7%).

### 3.3. Detectability Indexes

For all CT systems and all phantom diameters, the d’ values decreased as the energy level increased (Figure 5). From 40 to 70 keV, the d’ values decreased on average for all phantom diameters by −21.7% ± 6.5% for SFCT-1st, −26.0% ± 2.5% for SFCT-2nd, and −28.3% ± 0.7% for DSCT.

For all CT systems and all energy levels, the d’ values decreased as the phantoms’ diameter increased. From the diameters of 26 cm to 31 cm, the d’ values decreased on average for all energy levels by −30.5% ± 1.6% for SFCT-1st, −29.8% ± 1.3% for SFCT-2nd, and −25.3% ± 0.4% for DSCT and from 31 cm to 36 cm, by −31.5% ± 3.3%, −24.4% ± 0.8%, and −20.1% ± 0.2%, respectively.

Whatever the phantom diameter and energy level, the lowest d’ values were found with SFCT-1st. For all energy levels, the d’ values were lower with DSCT than with SFCT-2nd for the 26 cm diameter phantom (−6.2% ± 0.7%), similar for the 31 cm diameter phantom (−0.3% ± 1.9%), and higher for the 36 cm diameter phantom (5.4% ± 2.7%).

## 4. Discussion

In the present study, for the first time, the performances of three dual-energy CT (DECT) platforms were compared according to the diameter of the phantoms. To achieve this goal, a task-based image quality assessment of the low-energy-level virtual monoenergetic images was performed. The results showed that, despite increasing the dose level, the greater the phantom’s diameter, the greater the image noise and degradation of the spatial resolution, noise texture and detectability. We also found that the lower keV led to an increase in contrast and detectability, despite an increase in noise magnitude and a degradation in noise texture and spatial resolution. Finally, the highest detectability was found with the SFCT-2nd platform for the smallest phantom diameter and, for the dual-source CT platform, with the largest diameter phantom.

The outcomes of the NPS confirmed that the noise magnitude increased as the phantoms’ diameter increased [20] and the energy levels decreased [16,17,19]. For all diameters of the phantoms, the noise magnitude values were higher with SFCT-1st than with SFCT-2nd and DSCT. Similar outcomes were found between SFCT-1st and SFCT-2nd using AuSn120 kVp using another phantom [19]. The noise magnitude values were higher with DSCT than with SFCT-2nd for all diameters, except for the 36 cm diameter phantom. Regarding noise texture, the f_av_ values increased as the energy levels increased and as the phantom’s diameter decreased. For all diameters of the phantoms and for all keVs, the best noise texture was found with DSCT, resulting in “fine” noise. Similar noise texture was found with both SFCTs for the 26 cm and 31 cm diameters, but the highest f_av_ values were found with the SFCT-2nd for the 36 cm diameter phantom. These results are consistent with previously published studies on the SFCT-1st which used a different image quality phantom and a different reconstruction kernel (Br40) [16,17] and on DSCT [20]. We also found the presence of two peaks on some NPS curves for all DECT platforms. For all phantom diameters, these two peaks were present on all the NPS curves of both SFCT platforms at 40, 50, and 60 keV. For these two platforms, the magnitude of the NPS peak at low frequencies was higher than the NPS peak’s magnitude at high frequencies, where the lower the keV, the greater the phantom diameter. This phenomenon was more pronounced with the SFCT-1st than with the SFCT-2nd. With DSCT, these two peaks were present at 40 keV for the three phantom diameters and at 50 and 60 keV only for the 36 cm diameter phantom. In addition, the magnitude of the two NPS peaks was similar for both spatial frequencies. The presence of these two NPS peaks had been found in previous studies [16,17,19] and may be related to the distortions in the images or artifacts on the VMIs at the lowest energy levels for these platforms.

The spatial resolution outcomes confirmed that, for the iodine insert at 10 mg/mL, the TTF values at 50% decreased as the phantom’s diameter increased [20]. We found that the f_50_ values decreased as the energy levels decreased. Similar outcomes were found for SFCT-1st and DSCT [16,17,20]. However, Greffier et al. found that the f_50_ values were not changed from 40 to 70 keV for SFCT-1st and SFCT-2nd [19]. This study was performed with a different phantom and inserts containing lower iodine concentrations (e.g., 2 and 4 mg/mL). These results were directly related to the noise magnitude and the contrast variations of the insert used according to the phantom’s diameter and the energy levels. TTF was calculated using the circular edge technique [23], which generates an Edge Spread Function (ESF) from the difference in the HU values between the phantom’s background material and the insert. The ESF therefore depends on the noise conditions in the image, as well as on the differences in the HU values between the background and the insert. We found that, for the same contrast, the image noise increased as the phantom diameter increased, altering the ESF, and therefore, the resulting TTF. Furthermore, for the same phantom diameter, the reduction in keV resulted not only in an increase in image noise, but also in an increase in contrast. Our results seem to suggest that the impact of noise on the ESF was more marked than that of the contrast, since the f_50_ values tended to decrease as the low keVs decreased. Finally, based on these variations in contrast and image, the highest f_50_ values were found for DSCT, particularly for the diameters of 26 and 31 cm. With the SFCT platforms, the f_50_ values were higher with the SFCT-2nd than with the SFCT-1st for the 26 cm diameter phantom, and the opposite was true for the other diameters.

The results of the detectability index (d’) values calculated for the simulated lesions show that the d’ values peaked at 40 keV and decreased as the energy levels increased. Similar outcomes were obtained by Euler et al. for DSCT [20] for the same phantom and by Greffier et al. for SFCT-1st and DSCT [17] and for SFCT-2nd for other image quality phantoms [19]. The d’ values were also found to decrease as the phantom’s diameter increased. For all diameter phantoms and energy levels, the lowest detectability was found with the SFCT-1st platform. The highest detectability was found with the SFCT-2nd for the 26 cm diameter phantom and for DSCT for the 36 cm diameter phantom. Similar d’ values were found for DSCT and SFCT-2nd for the 31 cm diameter phantom. As the contrast values were set for each energy level, these outcomes were directly related to the noise magnitude outcomes. Indeed, unlike the TTF, the NPS curves are not normalized and the variations in noise magnitude have a strong impact on the detectability results.

The results of our study show that the spectral performance of the three DECT platforms understudy varied according to the phantoms’ diameters and the technology used. Although the dose level increased between the 26 and 36 cm diameter phantoms, increasing the phantom’s diameter was found to alter the detectability, spatial resolution, amplitude, and noise texture. Furthermore, different results could be obtained with the higher dose levels, particularly for the 36 cm diameter phantom. However, to represent clinical practice as closely as possible, the dose levels were defined from the dose levels of abdomen CT examinations performed on the three CT systems used for patients with BMIs similar to those extrapolated for each phantom’s diameter. Concerning the technology used, we were able to highlight the different results depending on the DECT platform used. Compared to the two other DECT platforms, the SFCT-1st platform, with a limited spectral separation, presented the worst detectability and noise magnitude results and was the most influenced by the increase in phantom diameter. The technological evolutions brought to the SFCT-2nd platform by thickening the Au and Sn filters for better spectral separation have led to better results than with the SFCT-1st platform [19]. These improvements have brought the detectability results closer to those obtained with DSCT and even better for the 26 cm diameter phantom. However, these results must also be interpreted with caution as, in this study, only one pair of kVps (100/Sn150 kVp) out of the five available was studied, and different results might be obtained with another pair of kVps. In all cases, the results of this study show that the best detectability was obtained at 40 keV for the three CT systems used and therefore suggest that the low keV VMIs can be used in clinical routine. In order to guarantee optimal image quality on the keV-based VMIs, special attention must be paid to the dose level used for overweight patients. The results obtained on the phantoms in this study must now be confirmed via specific clinical applications on patients.

This study has certain limitations. The acquisitions were performed with only one quantitative reconstruction kernel and only one level of the ADMIRE algorithm was used. Other parameter combinations may provide different outcomes. However, the acquisition and reconstruction parameters used in this study were those used in clinical routine on all DECT systems assessed. Also, we did not use a tube current modulation system on the three CT systems. This choice made to guarantee similar dose levels in each phantom diameter for each CT system assessed. Finally, only one task function was chosen to model the detection of the contrast-enhanced lesion, with the contrast defined according to the contrast value variations relative to the VMIs for the one iodine insert available in the Mercury v4.0 phantom. This method was chosen to take into account the variations in the contrast of the simulated lesion according to the energy level. However, a clinical contrast-enhanced study on the lesions of a known specific size should really be carried out now to confirm our results.

## 5. Conclusions

The outcomes of the present study confirmed that the spectral performance depends on the DECT platform and the phantom’s diameter. Increasing the phantom’s diameter increased the image noise and degraded the spatial resolution, noise texture, and detectability, despite the increase in dose level. Changing the thickness of the tin and gold filters on the SFCT-2nd platform has led to a lower noise magnitude and better detectability than with the SFCT-1st. Compared with the DSCT platform, the SFCT-2nd had the lowest noise magnitude and highest detectability for the smallest phantom diameter, and the opposite was true for the largest diameter. Further patient studies are now required for specific clinical applications to confirm the results obtained on the phantoms in this study.

## Figures and Tables

**Figure 1 diagnostics-13-03039-f001:**
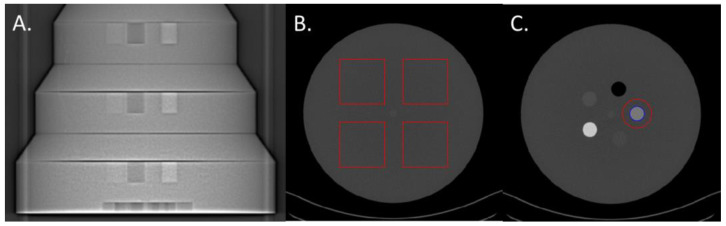
(**A**) Front X-ray image of the 26, 31, and 36 cm diameter sections of the Mercury v4.0 phantom. (**B**) Example of the four regions of interest (ROIs, in red) placed in the homogenous part of the 31cm diameter section and used on VMIs to assess the noise power spectrum (NPS). (**C**) Example of the ROI placed around the iodine insert of 10 mg/mL (in blue) and the phantom’s background material (in red) and used to compute the task-based transfer function (TTF).

**Figure 2 diagnostics-13-03039-f002:**
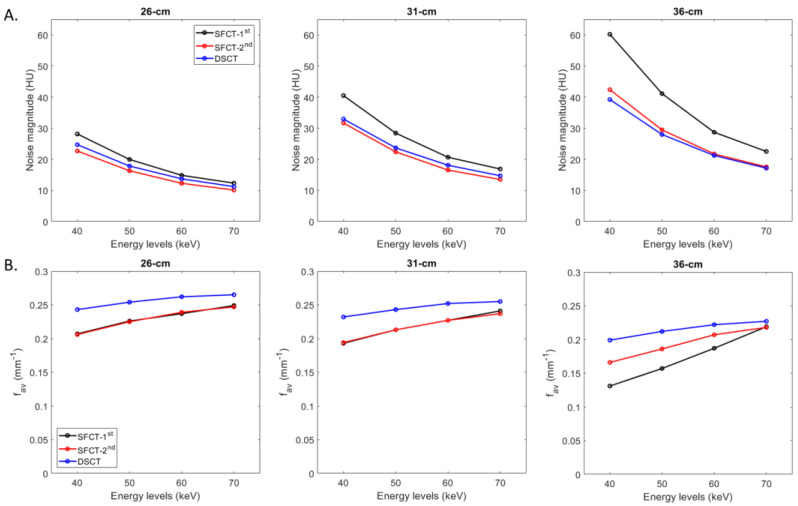
Values of noise magnitude and average Noise Power Spectrum spatial frequencies (f_av_) obtained for the two generations of split-filter CT platforms (SFCT) and the dual-source CT platform (DSCT) according to phantom diameter and energy level (keV).

**Figure 3 diagnostics-13-03039-f003:**
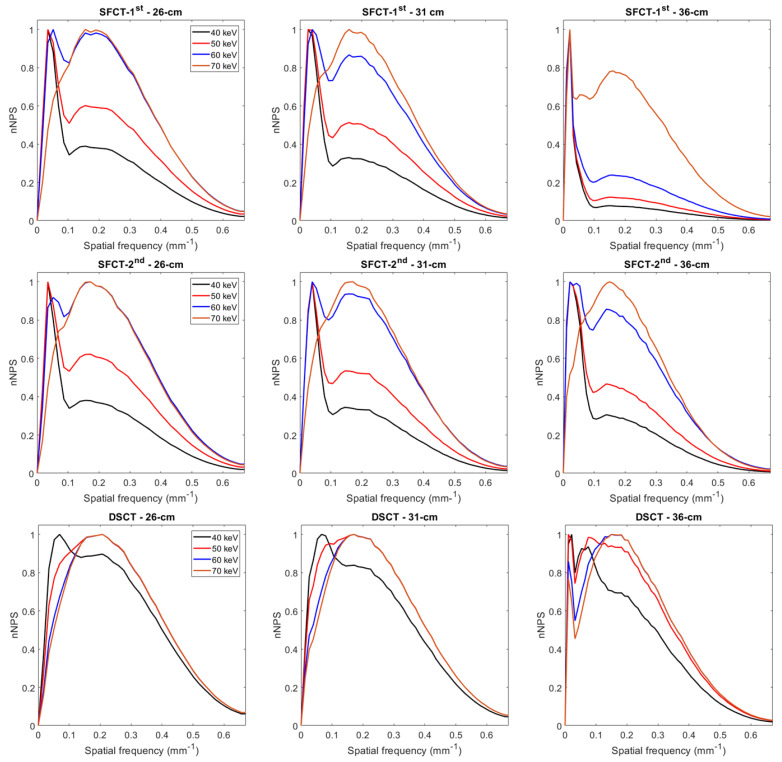
Normalized noise power spectrum (nNPS) curves obtained for the two generations of split-filter CT platforms (SFCT) and the dual-source CT platform (DSCT) according to phantom diameter and energy level (keV).

**Figure 4 diagnostics-13-03039-f004:**
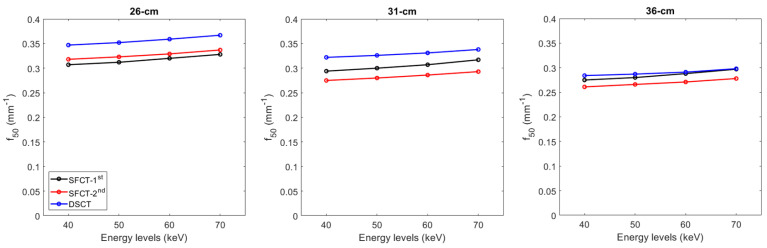
Values of task-based transfer function at 50% (f_50_) obtained for the two generations of split-filter CT platforms (SFCT) and the dual-source CT platform (DSCT) according to phantom diameter and energy level (keV).

**Figure 5 diagnostics-13-03039-f005:**
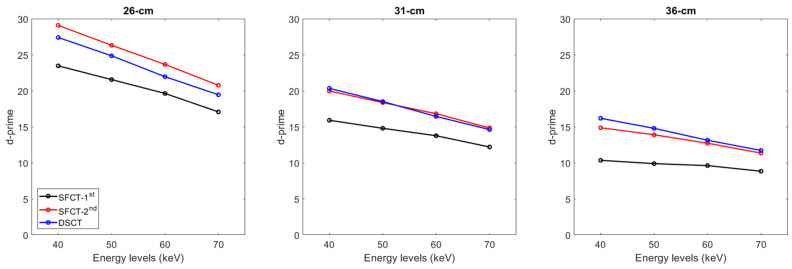
Detectability index (d′) values for the simulated lesion obtained for the two generations of split-filter CT platforms (SFCT) and the dual-source CT platform (DSCT) according to phantom diameter and energy level (keV).

**Table 1 diagnostics-13-03039-t001:** Acquisition parameters used for the three CT systems.

CT Scan Model		SOMATOM Edge	SOMATOM X.Cite	SOMATOM Force
SFCT system generation		1st generation	2nd generation	-
AuSn filter thickness	Au (mm)	0.05	0.07	-
	Sn (mm)	0.60	0.70	-
Tube voltage (kVp)		AuSn120	AuSn120	100/Sn150
Pitch factor		0.3	0.3	0.6
Rotation time (rot/s)		0.33	0.3	0.5
Beam collimation (mm)		38.4	38.4	38.4
CTDI_vol_ (mGy)	26 cm diameter 31 cm diameter 36 cm diameter	6.89 9.10 11.99	6.95 9.18 12.15	6.86 9.10 12.04

Footnotes: AuSn: gold and tin filters; CTDI_vol_: volume CT dose index; SFCT: split-filter CT.

**Table 2 diagnostics-13-03039-t002:** Values of noise power spectrum peak(s) and their respective spatial frequencies (f_peak_) obtained for all energy levels (keV) on the three CT systems.

		NPS Peak (HU^2^·mm^2^)				f_peak_ (mm^−1^)			
	Diameter	40 keV	50 keV	60 keV	70 keV	40 keV	50 keV	60 keV	70 keV
SFCT-1st	26 cm	3514/1369	1157/696	414/407	288	0.03/0.16	0.03/0.16	0.05/0.16	0.16
	31 cm	8675/2862	2805/1439	933/808	561	0.03/0.16	0.03/0.16	0.05/0.16	0.16
	36 cm	80824/6322	24714/3036	6606/1574	1350/1057	0.02/0.15	0.02/0.15	0.02/0.16	0.02/0.16
SFCT-2nd	26 cm	2427/922	781/486	261/285	199	0.04/0.16	0.04/0.17	0.05/0.17	0.17
	31 cm	5370/1845	1753/936	566/531	370	0.04/0.15	0.04/0.15	0.04/0.16	0.17
	36 cm	12320/3755	3978/1853	1198/1026	719	0.02/0.14	0.02/0.14	0.02/0.14	0.15
DSCT	26 cm	1131/1015	536	320	214	0.07/0.21	0.21	0.21	0.21
	31 cm	2232/1873	981	579	384	0.07/0.17	0.17	0.17	0.17
	36 cm	4541/4248	1757/1729	826/962	632	0.02/0.07	0.01/0.07	0.01/0.15	0.15

Footnotes: DSCT; Dual-source CT; f_peak_: spatial frequencies of the noise power spectrum peak(s); SFCT-1st: first generation of split-filter CT; SFCT-2nd: second generation of split-filter CT.

## Data Availability

The data presented in this study are available from the corresponding author upon reasonable request.

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
