# Peer review of "Impact of Phantom Size on Low-Energy Virtual Monoenergetic Images of Three Dual-Energy CT Platforms"

_diagnostics, 2023, doi:10.3390/diagnostics13193039_

Round 1

Reviewer 1 Report

The comments are attached in PDF.

moderate editing required

Author Response

Reviewer 1:

Dear Editor and Dear Authors,

The topic of this paper is very interesting and valuable from a practical point of view, and thank you for your trust to send me the manuscript to review it.

This paper compares the quality of low-energy VMIs obtained with three DECT platforms according to phantom diameter. The experiments were conducted and the conclusions were drawn.

The comments and suggestions are below:

  1. The use of personal pronounses (we, our, etc.) is unusual for scientific English. The suggestion is to use Passive Voice instead in the whole paper to improve it.

Reply: Following the reviewer’s comment, the medical writer (English native speaker) has now revised the manuscript.

  1. All the abbreviations should be explained at the first appearance, started in abstract (i.e. VMI). On the other hand, the abstract contains many abbreviations and it could be very repulsive for readers. Also, the abstract should reflect the research and paper itself to attract the readers. I think the abstract should be rewritten with more concise wording to highlight the main aspects of the paper. I think the numerical data should be placed in the result discussion section.

Reply: Following the reviewer’s comment, we have now explained all the abbreviations upon first appearance in the Abstract and in the whole manuscript.

Concerning the reviewer's comment about the abstract, as we have not had similar comments from the other two reviewers regarding the abstract, we prefer not to rewrite it.

  1. The Introduction is satisfactory, however in the end of the section a brief summary of the paper should be given.

Reply: To take the reviewer’s comment into account, we have now rephrased the last paragraph of the Introduction section as follows:

"The purpose of our study was to compare the spectral performances of three DECT platforms according to phantom diameter. To achieve this goal the spectral performances of the three DECT platforms were assessed by carrying out a task-based image quality assessment on VMIs at low energy levels using the three largest sections of an image quality phantom."

  1. Where is the Related Work section? I think authors should discuss other works from the literature in order to make an appropriate proposal of their research. This part is mandatory for all scientific papers.

Reply: We thank the reviewer for this pertinent comment. However, in the penultimate paragraph of the introduction, we have discussed other work from the literature and highlighted the limitations of those studies, which led to the present study.

We have also added a recently published article comparing the spectral performance of the two generations of SFCTs on a phantom (with a single morphology).

"Two studies have highlighted differences in spectral performance between SFCT-1st and DSCT [16,17]. Using two X-ray tubes provides good spectral separation for DSCT. Otherwise, with SFCT, spectral separation is more limited, due to the overlap between the low- and high-energy spectra generated by the two filters. These differences in spectral performance translate into greater image noise and poorer lesion detectability on the virtual monoenergetic images (VMIs) at low energy levels [16,17]. Only one study has evaluated the impact of modifying the thickness of the two filters on the spectral performance of SFCT-2nd compared with SFCT-1st [19]. The authors demonstrated that modifications in the thickness to the gold and tin filters of SFCT-2nd improved spectral separation and also spectral performance (especially, lowest values of noise magnitude, highest values of detectability indexes) compared to SFCT-1st for low-keV VMIs. All three studies were carried out using a phantom with a fixed morphology. In addition, spectral image quality is also influenced by patient morphology and the associated dose level. Euler et al. evaluated the impact of phantom diameter and dose level on the spectral performance of the DSCT platform [20]. However, no similar studies have been conducted on the two generations of SFCT platforms. "

  1. I think the method section should contain more details. The current explanations are superficial. What image enhancing methods (that are included in the CT software) were used during the research?

Reply: We thank the reviewer for this comment. However, we have not received similar comments from the other 2 reviewers. We consider that we have provided sufficient detail in the "Materials and Methods" section to make it understandable and possibly reproducible for readers. We therefore prefer to leave this section unchanged.

We also find the reviewer's question: "What image enhancing methods (that are included in the CT software) were used during the research?" difficult to understand.

No “image enhancing methods” were used in this study. As stated in our article, VMIs were generated using the usual parameters on Syngo.via software (VB60A_HF03) with the specific Monoenergetic Plus application. The VMIs were then analyzed using the iQMetrix-CT software, whose operating principle and features have already been described by Greffier et al. (doi:10.1016/j.diii.2022.05.007).

  1. The experiments and results are satisfactory explained. Are there any image results to include?

Reply: We thank the reviewer for his/her comment. In the “Supplementary file” we have added the axial images of the iodine insert obtained for the 31-cm diameter for the three DECT platforms and 4 energy levels.

  1. The Conclusion section is satisfactory.

Reply: We thank the reviewer for this positive comment.

  1. Are there any future works?

Reply: We thank the reviewer for this question and, as specified in the Discussion, we plan to perform a patient study to confirm the results found in this study. To take your comment into account we have added the following sentence in the conclusion “Further patient studies are now required for specific clinical applications to confirm the results obtained on phantoms in this study.

The paper is very interesting from a practical point of view, however it has drawbacks that should be corrected.

Recommendation: major revisions.

Reviewer 2 Report

The authors consider, there are no studies that have evaluated the impact of modifying the thickness of the two filters on the spectral performance of SFCT-2nd, especially compared with SFCT-1st and DSCT. Spectral image quality is influenced by patient morphology and the dose level.

There are not evaluations of the impact of phantom diameter and dose level on the spectral performance on the two generations of SFCT platforms. In this study, the authors compared the spectral performances of three DECT platforms according to phantom diameter. Spectral performances were assessed by carrying out a task-based image quality assessment on VMIs.

The authors presented performances of three dual-energy CT (DECT) platforms comparing them according to the diameter of phantoms. A task-based image quality assessment of low-energy-level virtual monoenergetic images was performed, showing that despite increasing the dose level, the greater the phantom’s diameter, the greater the image noise and degraded spatial resolution, noise texture, and detectability. Lower keV led to an increase in contrast and detectability, despite an increase in noise magnitude and a degradation in noise texture and spatial resolution. The highest detectability was found with the SFCT-2nd platform for the smallest phantom diameter and, for the dual-source CT platform, with the largest diameter phantom.

Comment: in opinion of this reviewer, the authors should highlight the principal contribution of their study contracting these ones against existed approaches.

Author Response

Reviewer 2:

The authors consider, there are no studies that have evaluated the impact of modifying the thickness of the two filters on the spectral performance of SFCT-2nd, especially compared with SFCT-1st and DSCT. Spectral image quality is influenced by patient morphology and the dose level.

There are not evaluations of the impact of phantom diameter and dose level on the spectral performance on the two generations of SFCT platforms. In this study, the authors compared the spectral performances of three DECT platforms according to phantom diameter. Spectral performances were assessed by carrying out a task-based image quality assessment on VMIs.

The authors presented performances of three dual-energy CT (DECT) platforms comparing them according to the diameter of phantoms. A task-based image quality assessment of low-energy-level virtual monoenergetic images was performed, showing that despite increasing the dose level, the greater the phantom’s diameter, the greater the image noise and degraded spatial resolution, noise texture, and detectability. Lower keV led to an increase in contrast and detectability, despite an increase in noise magnitude and a degradation in noise texture and spatial resolution. The highest detectability was found with the SFCT-2nd platform for the smallest phantom diameter and, for the dual-source CT platform, with the largest diameter phantom.

Comment: in opinion of this reviewer, the authors should highlight the principal contribution of their study contracting these ones against existed approaches.

Reply: We thank the reviewer for his/her comment. As stated in the Introduction section, we have outlined the novel aspects of this study, highlighting the limitations of existing studies and the absence of similar studies. In the Discussion section, we have compared the results found in the present study with the results of existing studies, specifying the differences between these studies and our study each time.

We have now modified the Introduction and Discussion to take into account your comments and those of the other two reviewers, and added a recent study comparing SFCT-1st and SFCT-2nd on a phantom with a unique morphology.

Reviewer 3 Report

In this study, the authors have compared the quality of low-energy virtual monoenergetic images obtained with three DECT platforms according to phantom diameter. Three sections of the Mercury Phantom 4.0 were scanned on two generations of split-filter CTs (SFCT-1st and SFCT-2nd) and on one dual-source CT (DSCT). The authors demonstrated that compared to SFCT-1st, SFCT-2nd exhibited lower noise magnitude and  higher detectability values. Compared with DSCT, SFCT-2nd had lower noise magnitude and higher detectability for the 26-cm but the opposite was true for the 36-cm.

As a general comment, this is a well performed study that addresses an important issue.

I suggest the following minor amendments

Index terms: Multidetector Computed tomography should read Multidetector computed tomography

 Abstract

The first sentence should start with “The purpose of this study was to compare

VMI should be spelt out

Introduction

It sounds quite weird to write “Numerous studies have highlighted differences in spectral performance between SFCT-1st and DSCT [16,17].” when only two references are given.

Materials and Methods

The authors should indicate the city and the country of the manufacturers

A foot note should be added to table 1 to explain all acronyms/abreviations used in the Table.

The authors should indicate that results were reported as means ± standard deviations [SD) in a systematic fashion m ± x (SD) unit or %

Results

Foot notes should be added to tables to explain all acronyms/abreviations used in the Tables.

Author Response

Reviewer 3:

We would like to thank the reviewer for all his comments.

Index terms: Multidetector Computed tomography should read Multidetector computed tomography

Reply: This correction has now been made.

 Abstract

The first sentence should start with “The purpose of this study was to compare

Reply: This correction has now been made.

VMI should be spelt out

Reply: This correction has now been made.

Introduction

It sounds quite weird to write “Numerous studies have highlighted differences in spectral performance between SFCT-1st and DSCT [16,17].” when only two references are given.

Reply: Following the reviewer's most pertinent comment, we have now replaced "Numerous studies" by "Two studies".

Materials and Methods

The authors should indicate the city and the country of the manufacturers

Reply: This correction has now been made.

A foot note should be added to table 1 to explain all acronyms/abbreviations used in the Table.

Reply: This correction has now been made.

The authors should indicate that results were reported as means ± standard deviations [SD) in a systematic fashion m ± x (SD) unit or %

Reply: Following the reviewer's comment, we have now added the following sentence: "In the Results section, values are expressed as means ± standard deviations (SD)."

Results

Foot notes should be added to tables to explain all acronyms/abbreviations used in the Tables.

Reply: This correction has now been made in Table 2.

Round 2

Reviewer 1 Report

All the comments are answered and the paper has been improved. Thank you.

Recommendation: accept

minor editing

Author Response

Thank you very much for your review report